# Fatigue Is a Major Symptom at COVID-19 Hospitalization Follow-Up

**DOI:** 10.3390/jcm11092411

**Published:** 2022-04-25

**Authors:** Søren Sperling, Andreas Fløe, Steffen Leth, Charlotte Hyldgaard, Tina Gissel, Ayfer Topcu, Lars Kristensen, Lene Dahl Sønderskov, Johannes Martin Schmid, Søren Jensen-Fangel, Elisabeth Bendstrup

**Affiliations:** 1Department of Respiratory Diseases and Allergy, Aarhus University Hospital, 8200 Aarhus, Denmark; andriniel@rm.dk (A.F.); johaschm@rm.dk (J.M.S.); karbends@rm.dk (E.B.); 2Department of Clinical Medicine, Aarhus University, 8000 Aarhus, Denmark; stefleth@rm.dk (S.L.); soejense@rm.dk (S.J.-F.); 3Department of Infectious Diseases, Regional Hospital West Jutland, 7400 Goedstrup, Denmark; 4Department of Infectious Diseases, Aarhus University Hospital, 8200 Aarhus, Denmark; 5Diagnostic Centre, Silkeborg Regional Hospital, 8600 Silkeborg, Denmark; charhyld@rm.dk; 6Department of Respiratory Diseases, Viborg Regional Hospital, 8800 Viborg, Denmark; tina.n.gissel@midt.rm.dk; 7Department of Respiratory Diseases, Horsens Regional Hospital, 8700 Horsens, Denmark; ayfetopc@rm.dk; 8Department of Respiratory Diseases, Regional Hospital West Jutland, 7400 Goedstrup, Denmark; larskris@rm.dk; 9Department of Respiratory Diseases, Randers Regional Hospital, 8900 Randers, Denmark; lene.dahl@aarhus.rm.dk

**Keywords:** COVID-19, long COVID, fatigue, pulmonary function

## Abstract

Persistent symptoms after hospitalization with COVID-19 are common, but the frequency and severity of these symptoms are insufficiently understood. We aimed to describe symptoms and pulmonary function after hospitalization with COVID-19. Patients hospitalized with COVID-19 in Central Denmark Region were invited for follow-up 3 months after discharge. Clinical characteristics, patient reported outcomes (Fatigue Assessment Scale (FAS), anxiety and depression (HADS)), symptoms, pulmonary function test and 6-min walk test were collected. We included 218 patients (mean age 59.9 (95% CI: 58.2, 61.7), 59% males). Fatigue, dyspnea and impaired concentration were the most prevalent symptoms at follow-up. Using FAS, 47% reported mild-to-moderate fatigue and 18% severe fatigue. Mean HADS was 7.9 (95% CI: 6.9, 8.9). FAS was correlated to HADS (β = 0.52 (95% CI: 0.44, 0.59, *p* < 0.001)). Mean DLCO was 80.4% (95% CI: 77.8, 83.0) and 45% had DLCO ˂ 80%. Mean DLCO was significantly reduced in patients treated in the ICU (70.46% (95% CI 65.13, 75.79)). The highest FAS and HADS were seen in patients with the shortest period of hospitalization (2.1 days (95% CI: 1.4, 2.7)) with no need for oxygen. In conclusion, fatigue is a common symptom after hospitalization for COVID-19 and ICU treatment is associated to decreased diffusion capacity.

## 1. Introduction

In December 2019, a new severe acute respiratory syndrome coronavirus 2 (SARS-CoV-2) emerged in Wuhan, China, causing Coronavirus disease 2019 (COVID-19) [1]. At this point, more than 452 million COVID-19 cases and 6 million deaths have been confirmed worldwide. COVID-19 is a heterogeneous disease affecting the respiratory system, with fever, fatigue and cough as the most prevalent symptoms [2]. The disease course spans from asymptomatic infection to severe respiratory and multiorgan failure and subsequent death. Approximately 95% of patients experience a mild or moderate course of disease [3,4]. Elderly patients, especially those with one or more comorbidities, and non-vaccinated individuals are at risk of severe disease and fatal outcome [5,6,7].

At the beginning of the first pandemic wave, reports suggested an increase in persistent multiorgan symptoms months after acute illness; this was later defined as long Covid if symptoms persisted for more than three months [8]. The most prevalent symptoms of long Covid are fatigue (approx. 70%) and dyspnea (approx. 50%) reported in patients months after discharge [9,10,11]. Furthermore, several studies have found radiological abnormalities [12], mildly reduced diffusion capacity for carbon monoxide (DLCO) [13] and relatively preserved dynamic lung volumes, which contrast patients’ experience of debilitating dyspnea months after admission [14].

Recent evidence on long Covid suggests that females and patients with a medical history of mental disease (anxiety or depression) are at increased risk of prolonged fatigue [15]. One study [16] found that females were at increased risk of mental disease at follow-up, including sleep disorder, anxiety and depression. Studies found that patients with severe disease [17], and multiple symptoms in the acute phase of COVID-19 [18] were at higher risk of developing prolonged fatigue and long Covid. However, Shang et al. [16] found no association between disease severity and risk of long-term fatigue. So far, the pathophysiological background for long Covid remains largely unknown.

The aim of the present study was to systematically describe the frequency and form of ongoing long-term respiratory symptoms reported after hospitalization for COVID-19 and relate these findings to physiological measures and patient reported outcome measures (PROMs).

## 2. Materials and Methods

### 2.1. Study Participants

Patients at 18 years or older hospitalized for COVID-19 with a confirmed SARS-CoV-2 infection by positive polymerase chain reaction (PCR) testing were referred to a clinical follow-up visit three months after hospitalization at six hospitals in the Central Denmark Region with a total population of approx. 1.3 million. At the clinical follow-up visit, patients were invited to participate in the study.

Patients were included after giving written informed consent. The study was approved by The Central Denmark Region Committee on Health Research Ethics (file number 1-10-72-130-20) and The Danish Data Protection Agency (1-16-02-203-20). The trial is registered in clinicaltrials.gov (ID NCT04401163).

### 2.2. Clinical Data

Data were obtained from patient records on demographics, time from onset of symptoms to admission, length of hospitalization and treatment including oxygen therapy, nasal high flow oxygen (NHF), ICU stay and/or, ventilator treatment, pharmacological treatment as well as comorbidity and smoking status. Administration of Remdesivir and Dexamethasone (RaD) was registered. Treatment with RaD was implemented and changed during the inclusion period for the included subjects, and administration pattern for these drugs thus reflects changing treatment recommendations.

At follow-up, a structured interview guide was used to register information on respiratory symptoms and duration together with physiological tests and PROMs.

### 2.3. Pulmonary Function and Six-Minute Walk Test

At follow-up, patients completed a pulmonary function test (PFT) including flow-volume curves and body plethysmography with registration of forced expiratory volume in one second (FEV1), forced vital capacity (FVC), total lung capacity (TLC), residual volume (RV) and diffusion capacity (DLCO) for carbon monoxide (MasterScreen PFT; Jaeger, Germany). PFT was conducted according to European Respiratory Society and American Thoracic Society (ATS) guidelines and stated according to reference values [19,20,21].

At follow-up, exercise capacity was assessed by a six-minute walk test (6MWT) [22]. Reference equations were used to calculate predicted 6MWT distance (6MWTD) [23]. Patient-reported dyspnea before, during and after the 6MWT was measured using the Borg scale [24]. Peripheral arterial oxygen saturation was measured using a pulse oximeter placed on the index finger, and pre-test oxygen saturation as well as maximum desaturation was recorded.

### 2.4. Patient Reported Outcome Measures

The Fatigue Assessment Scale (FAS) is a self-reported 10-item questionnaire validated in patients with sarcoidosis [25]. The total score ranges from 10 to 50. A FAS score between 0–21 indicates no fatigue, 22–34 indicates mild-to-moderate fatigue and ≥35 indicates severe fatigue [26].

The MRC scale contains five statements that score the degree of disability associated with dyspnea. A score of 1 indicates no disability, and a score of 5 indicates severe incapacity [27].

The Hospital Anxiety and Depression Scale (HADS) is a 14-item scale assessing depression and anxiety with emphasis on minimizing the impact of physical illness on the total score. The items are rated on a 4-point severity scale. HADS consists of two scales, one for depression, HADS-D, and one for anxiety, HADS-A. Based on previous studies, a cut-off ≥8 was used to identify cases [28,29].

### 2.5. Statistics

Continuous data are presented as means with 95% confidence intervals (95% CI). Normal distribution of data was assessed using histograms and Q-Q plots. Categorical variables are reported as proportions of the total population.

The impact of clinical variables on disease severity was tested using multivariable linear regression models for continuous data yielding a regression coefficient (β) describing the differences between groups. Comparison of binary outcome variables between treatment groups were made using multivariable logistic regression yielding odds ratio (OR). A backward stepwise selection was used to select variables associated with outcomes. Variables age, sex and BMI were included in the multivariable model, all with a *p*-value < 0.2.

Patients who did not receive oxygen during hospitalization were the reference group in the statistical analysis. A *p*-value < 0.05 was considered statistically significant.

Stata (version 17.0, StataCorp, College Station, TX, USA) was used for statistical analysis.

## 3. Results

### 3.1. Baseline Data during Hospitalization

A total of 218 patients were included (mean age 59.9 (95% CI: 58.2, 61.7)), 128 (59%) males. Mean body mass index (BMI) was 29.2 (95% CI: 28.5, 29.9) and 106 (54%) had never smoked. The most frequent comorbidities were hypertension, asthma and diabetes. Table 1 presents basic demographic characteristics of participants. See Appendix A, for additional details.

During admission, 57 (26%) patients received no oxygen therapy, 120 (55%) received oxygen at the ward and 41 (19%) were treated at the ICU. A total of 22 (10%) patients received ventilator treatment, and one patient was treated with extracorporeal membrane oxygenation (ECMO).

In a multivariable model, patients in the ICU were older (mean age 62.3 vs. 54.0, *p* = 0.057) had a higher BMI (30.4 vs. 28.2, *p* = 0.067), and were more often males (80% vs. 42%, *p* = 0.005) compared to patients who did not need oxygen therapy.

Mean time from onset of symptoms to admission was 8.4 days (95% CI: 7.4, 9.4), and did not differ between patients receiving oxygen therapy or not, or whether treatment in the ICU was needed or not. Overall, mean length of hospital stay was 9.3 days (95% CI: 7.7, 10.9); longest length of stay (26.4 days, 95% CI: 21.3, 31.4) was seen in patients treated in the ICU.

### 3.2. Follow-Up

Mean time from hospitalization to follow-up was 127.7 days (95% CI: 122.2, 133.1). The most prevalent symptoms at follow-up were fatigue (61%), dyspnea (55%) and impaired concentration (34%). Overall, 86% of all patients reported at least one symptom. Symptoms reported at follow-up did not differ significantly between groups (Figure 1). See Appendix A, for additional symptoms at follow-up. Table 2 presents patient characteristics at follow-up.

### 3.3. Pulmonary Function Test

At follow-up, mean FEV1 was 2.9 l (95% CI: 2.8, 3.0) and FEV1 percent predicted (FEV1%) was 98.2% (95% CI: 95.5, 100.9). A total of 27 patients (13%) had FEV1 ˂ 80% predicted, and 11 patients (5%) FEV1 ˂ 60% predicted. Mean TLC and RV percent predicted were within the normal range. Mean DLCO was 80.4% (95% CI: 77.8, 83.0) predicted (DLCO%). DLCO% was ˂80% predicted in 96 (45%) patients and ˂60% in 35 patients (16%).

FEV1, FVC, DLCO, TLC and RV were all significantly lower in patients admitted to the ICU compared to patients not needing oxygen therapy. However, only DLCO was decreased below the lower limit of normal (Table 2).

### 3.4. Six-Minute Walk Test (6MWT)

Mean 6MWTD at follow-up was 486.9 m (95% CI: 471.9, 501.9). The predicted walk distance was lower in males (84.7%, 95% CI: 81.6, 87.8) compared to females (95.9%, 95% CI: 91.4, 100.6), (*p* < 0.001). Mean desaturation was 2.9 percentage points (95% CI: 2.4, 3.4). A decrease in oxygen saturation of more than 4 percentage points was observed in 53 (27%) patients. Patients admitted to the ICU desaturated significantly more often ≥4 percentage points and below 92% compared to patients not needing oxygen therapy (Table 2).

### 3.5. Patient Reported Outcome Measures

#### 3.5.1. FAS

Most patients experienced significant fatigue; 88 patients (47%) had mild-to-moderate fatigue indicated by a FAS score of 22–34 and 34 patients (18%) had severe fatigue indicated by a FAS score ≥ 35. FAS scores were significantly higher in females compared to males (*p* = 0.02).

The highest FAS score was found in younger patients and patients who did not receive oxygen therapy during hospitalization (Figure 2, panel A and Table 2). Severe fatigue (FAS score ≥ 35) was more often reported in these groups (Figure 2, panel B and Table 2).

#### 3.5.2. HADS

Mean HADS was 7.9 (95% CI: 6.9, 8.9). The HADS-A score was higher than HADS-D in all subgroups. The highest HADS-scores were found in younger patients and patients who did not receive oxygen therapy during hospitalization (Figure 3 and Table 2).

#### 3.5.3. MRC

Overall, mean MRC score was 1.9 (95% CI: 1.8, 2.1). Patients treated in the ICU had the highest MRC score (*p* = 0.08) and the highest percentage (*n* = 12, 31%) of patients with MRC scores ≥ 3 was seen among patients treated in the ICU.

A total of 48 patients had MRC scores ≥ 3 of whom 31 (65%) had a FEV1% within the normal range and 38 (79%) had a FVC percent predicted (FVC%) within the normal range. MRC scores ≥ 3 were reported by 19 (40%) patients with DLCO% within the normal range.

#### 3.5.4. Relationship between FAS and Other Parameters

A positive correlation between FAS and HADS total score was found (β = 0.52 (95% CI: 0.44, 0.59, *p* < 0.001)) indicating an increase in FAS of 1 was associated to an increase in HADS score of 0.52. In addition, a negative correlation was found between FAS and 6MWTD (−1.92 (−3.62, −0.22), *p* = 0.03) indicating patients with fatigue had a shorter walking distance. FAS was not related to DLCO, anticoagulant treatment or smoking status.

### 3.6. Remdesivir and Dexamethasone Subgroup Analysis

Patients receiving RaD were younger (*p* = 0.01) and had more comorbidities (1.8 (95% CI: 1.6, 2.0)) compared to patients not receiving RaD (1.1 (95% CI: 0.8, 1.4), *p* < 0.001). Length of hospital stay was non-significantly shorter in the group receiving RaD with a mean of 9.2 days (95%CI: 7.1, 11.4) compared to a mean of 12.3 days (95% CI: 8.8, 15.7) in the group not receiving RaD (*p* = 0.07).

Patients receiving RaD reported a higher HADS score (8.8 vs. 5.7, *p* = 0.008) and higher scores in both depression and anxiety subscales. However, a higher number of cases with a score ≥ 8 was not found in patients treated with RaD.

Pulmonary function test, 6MWTD, FAS and MRC were similar in the two groups. Patients treated with RaD had an increased risk of desaturation during 6MWT at follow-up (β 1.77, *p* = 0.02) and reported a higher change in Borg Scale score (β 1.16, *p* = 0.04). Table 3 shows patient characteristics stratified for treatment with RaD. See Appendix A, for additional details.

## 4. Discussion

Fatigue was a major symptom following hospitalization for PCR-proven COVID-19 infection at six hospitals in Central Denmark Region and the incidence at follow-up was in line with several previous studies reporting fatigue in 52–63% of participants [8,15,30]. Other common symptoms at follow-up were dyspnea (55%), impaired concentration (34%) and muscle pain (30%). The FAS score was increased in the majority of patients and the highest fatigue scores were found in younger patients, females and patients who had not received oxygen therapy. Mean DLCO and lung volumes were normal, although in 45% of patients the DLCO% was ˂80%. Patients treated in the ICU had the lowest DLCO and desaturated most often ≥4 percentage points during the 6MWT.

Dennis et al. and Goërtz et al. reported an incidence of fatigue of more than 95%. However, they used patient self-reported fatigue and did not report fatigue measured by standardized PROMs such as the FAS score [30,31]. Moreover, the study populations in these two studies included younger patients (median age 44 years and 47 years, respectively) and the majority were not hospitalized [32,33]. Conversely, other studies have reported a lower incidence of fatigue between 16% and 39% [34,35]. In the latter studies, patients had fewer comorbidities, which may explain the lower incidence of fatigue. Our study is the first to document a high incidence of fatigue among patients with a normal PFT.

Using validated PROMs, we found a high incidence of fatigue (65%), anxiety (23%) and depression (16%) in our cohort. Surprisingly, the highest fatigue and anxiety scores were found in younger patients hospitalized for the shortest period (2.1 days (95% CI: 1.4, 2.7)) without the need for oxygen therapy and with a correlation between fatigue and HADS. This is in keeping with a previous study reporting that fatigue was independent of disease severity and most pronounced in females [15]. Several mechanisms of the SARS-CoV-2 virus entering and affecting the central nervous system have been proposed and may explain the high incidence of neurocognitive deficits such as fatigue, concentration difficulty and dysgeusia at follow-up [36,37]. Furthermore, it may be speculated that external factors during a pandemic, e.g., physical inactivity during lockdown and emotional stress and concern could be a significant contributor to fatigue and reduced exercise capacity in long Covid [38,39].

In our cohort, dyspnea was reported by more than 50% of participants, and surprisingly, dyspnea was also a common symptom in the subgroup of patients with a normal FEV1 (52%) and DLCO (47%). Substantial dyspnea indicated by an MRC score ≥ 3 was frequent in patients with a normal FEV1% (65%), FVC% (79%) and DLCO% (40%) indicating that extrapulmonary factors most likely contribute to a high MRC score. We did not find any association between fatigue and dyspnea. Dysfunctional breathing has been suggested as a possible explanation for chronic dyspnea and exercise intolerance among some patients with long covid [31,40].

The 6MWTD was significantly shorter in patients who had received oxygen therapy or had been admitted to the ICU during hospitalization for COVID-19, and a shorter 6MWTD was related to an increased FAS, probably reflecting physical inactivity as discussed above. In addition, patients admitted to the ICU more often desaturated ≥4 percentage points and below 92%. These findings are in contrast to previous reports by Huang et al. and Van Den Borst et al., who both reported an equal walk distance across patients with mild, severe and critical disease [30,41]. Our findings correlate with the reduced DLCO in patients treated in ICU but did not correlate with fatigue. This may reflect that the lung parenchyma or ventilator-induced lung injury may have affected patients with severe respiratory failure during COVID-19 for several months after recovery, or it may represent pre-existing lung affection in patients who developed respiratory failure. It also indicates that post-covid dyspnea may have different physiological correlates in patients with mild and severe COVID-19, as self-reported dyspnea and high MRC scores were also highly prevalent in patients with mild disease and normal gas transfer. This finding underlines that dyspnea is a complex symptom that may involve several other mechanisms than those of the respiratory system.

Treatment with RaD was introduced in May 2020, which was during the inclusion period of this study. A non-significant reduction in length of admission and number of ICU admissions were found in patients receiving RaD, supporting the non-significant findings on Remdesivir by the WHO Solidarity Trial [42]. The use of RaD did not impact on long Covid concerning the burden of fatigue and other symptoms, although higher HADS and HADS-D scores were found. There was no increase in the number of cases with a score ≥ 8. The incidence of ICU admission was significantly reduced in patients receiving RaD (*p* = 0.02). One explanation to this finding could be that patients receiving RaD were significantly younger (*p* = 0.01). Only one study by Boglione et al. has previously investigated long Covid among patients treated with RaD. In contrast to our findings, Boglione et al. demonstrated reduced functional impairment at follow-up among patients treated with RaD using The Post-COVID-19 Functional Status scale [43,44]. Boglione et al. in their cohort found fatigue in 47.9%, myalgias/arthralgias in 40% and dyspnea in 50.8% at follow-up, which was similar to results in patients treated in the ICU. The study by Boglione et al. as well as the present study are observational and were not designed to evaluate the impact of RaD on long Covid; the results should thus be interpreted with caution. Other factors may impact on the results, as RaD was introduced concomitantly with changes of non-pharmacologic treatment and admission criteria to hospitals and to the ICU. Further studies are thus needed to clarify if RaD had a direct impact on long Covid.

Our study has some limitations. First, this observational study holds a risk of sampling bias, as lack of referral of patients with milder symptoms, or a higher willingness to participate in the study among patients experiencing debilitating symptoms, may have affected our results; also, we did not have access to information of the patients who did not participate. Second, the lack of pre-hospitalization baseline data makes it difficult to estimate the direct impact of COVID-19. However, we found that especially fatigue was overrepresented in younger persons with mild disease, making it less likely that this is a result of pre-existing conditions. Third, we cannot conclude if the results presented here are specific to COVID-19 or if they reflect a general tendency after a respiratory viral infection. Further studies including control groups representing healthy individuals and patients recovering from other infectious disease, e.g., influenza, pneumonia and common colds, would significantly increase our knowledge in this field.

However, our study also has several strengths. One of the strengths is the multicenter prospective design with patients included systematically during the first two waves of the pandemic. The study includes patients with different disease courses during hospitalization and highlights important PROMs and objective measures.

## 5. Conclusions

This study demonstrates that significant fatigue is common after hospitalization for Covid19 and is associated with younger age, shorter hospital stay and preserved pulmonary function. Contrary, desaturation during 6MWT and diffusion impairment mainly affected patients who had been treated in the ICU.

## Figures and Tables

**Figure 1 jcm-11-02411-f001:**
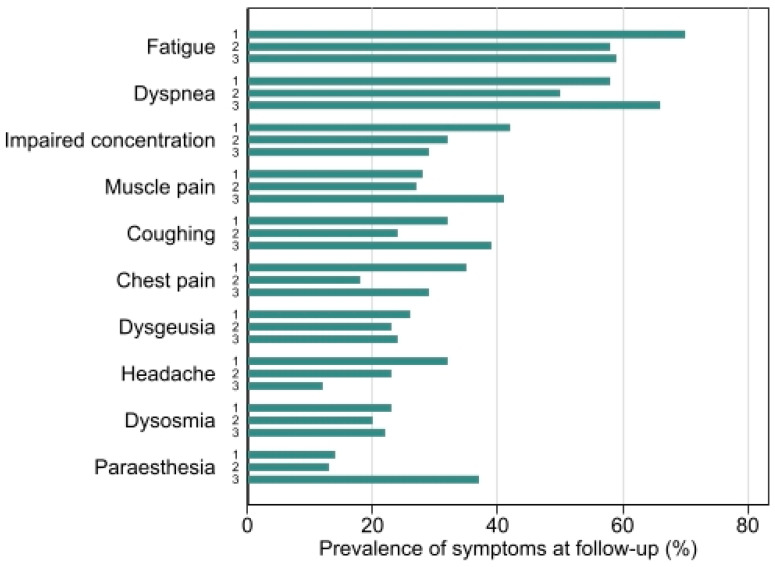
Prevalence of 10 most common symptoms at follow-up. Prevalence of each symptom is illustrated in 3 groups; (1) patients not requiring supplemental oxygen (2) patients requiring supplemental oxygen (3) patients admitted at ICU (3).

**Figure 2 jcm-11-02411-f002:**
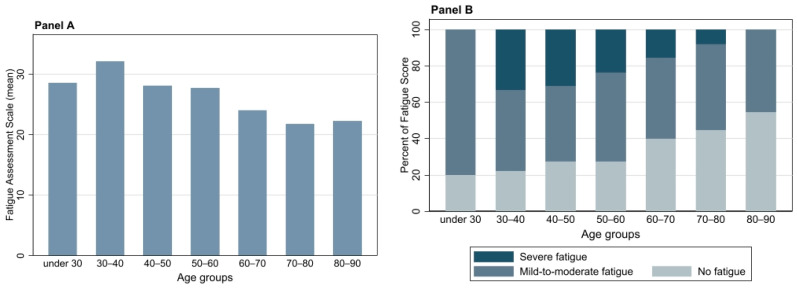
(**Panel A**) Graph bar showing the mean values of Fatigue Assessment Scale (FAS) by age groups. (**Panel B**) Graph bar showing the percent of patients with “Severe fatigue” (FAS score ≥ 35), “Mild-to-moderate fatigue (FAS score 22–34) and “No fatigue” (FAS score ≤ 21), by age groups.

**Figure 3 jcm-11-02411-f003:**
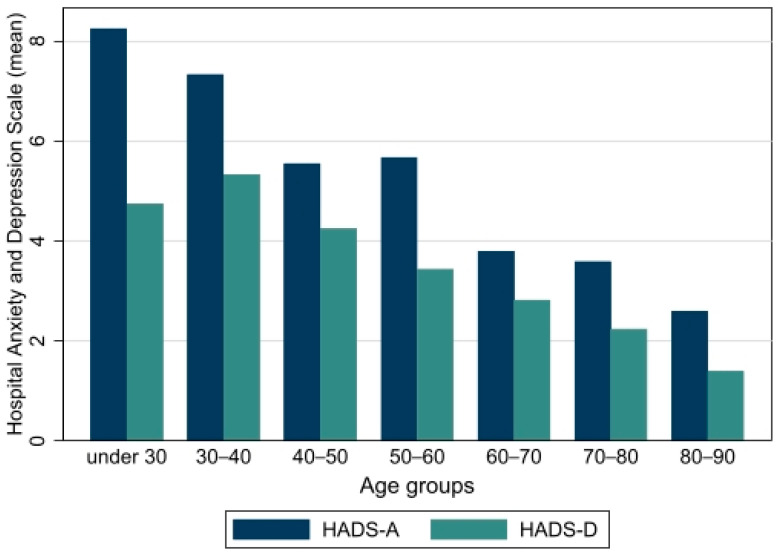
Mean scores from subscales on anxiety (HADS-A) and depression (HADS-D) in Hospital Anxiety and Depression Scale (HADS), by age groups. A score ≥ 8 identifies cases.

**Table 1 jcm-11-02411-t001:** Basic demographic characteristics during hospitalization.

	Overall. *n* = 218	Patients Not Requiring Supplemental Oxygen (Group 1) (*n* = 57)	Patients Requiring Supplemental Oxygen (Group 2) (*n* = 120)	Patients Admitted to ICU (Group 3) (*n* = 41)
Age (years)	59.94 (58.15, 61.73)	54.04 (50.36, 57.71)	62.3 (59.94, 64.66)	61.24 (57.58, 64.90)
Sex				
Male	128 (59%)	24 (42%)	71 (59%)	33 (80%)
Female	90 (41%)	33 (58%)	49 (41%)	8 (20%)
Body Mass Index	29.24 (28.51, 29.98)	28.21 (26.71, 29.71)	29.34 (28.31, 30.36)	30.37 (28.90, 31.84)
<18.5	1 (0%)	1 (2%)	0 (0%)	0 (0%)
18.5–24.9	44 (21%)	13 (24%)	25 (21%)	6 (15%)
25.0–29.9	90 (42%)	25 (45%)	50 (43%)	15 (37%)
≥30	78 (37%)	16 (29%)	42 (36%)	20 (49%)
Smoking status				
Ever	92 (46%)	23 (43%)	46 (43%)	23 (61%)
Never	106 (54%)	31(57%)	60 (57%)	15 (39%)
Comorbidities *				
Hypertension	71 (33%)	11 (19%)	41 (34%)	19 (46%)
Asthma	36 (17%)	8 (14%)	21 (18%)	7 (17%)
Diabetes	28 (13%)	4 (7%)	17 (14%)	7 (17%)
Malignancy	20 (9%)	7 (12%)	10 (8%)	3 (7%)
Chronic obstructive pulmonary disease (COPD)	18 (8%)	2 (4%)	11 (9%)	5 (12%)
Number of patients with more than 3 comorbidities	41 (19)	9 (16)	20 (17)	12 (29)
Time from symptom onset to admission (days)	8.37 (7.36, 9.37)	8.14 (6.20, 10.08)	8.42 (6.88, 9.95)	8.53 (7.24, 9.81)
Length of hospital stay (days)	9.29 (7.74, 10.86)	2.05 (1.37, 2.74)	6.85 (5.93, 7.77)	26.37 (21.30, 31.43)
Treatment during hospitalization				
Remdesivir	92 (42%)	6 (11%)	69 (58%)	17 (43%)
Systemic corticosteroids	119 (55%)	11 (19%)	81 (68%)	27 (68%)
Anticoagulation	153 (71%)	19 (33%)	96 (80%)	38 (97%)
Intravenous immunoglobin	2 (1%)	1 (2%)	1 (1%)	0 (0%)

Data are *n* (%), mean (95% CI) * Five most frequent comorbidities.

**Table 2 jcm-11-02411-t002:** Characteristics for patients hospitalized for PCR-proven COVID-19 at follow-up.

	All Patients (*n* = 218)	Patients Not Requiring Supplemental Oxygen (Group 1) (*n* = 57)	Patients Requiring Supplemental Oxygen ** (Group 2) (*n* = 120)	Patients Admitted at ICU (Group 3) (*n* = 41)	Group 2 vs. Group 1 Multivariable Model	Group 3 vs. Group 1 Multivariable Model
Time from discharge to follow-up (days)	127.65 (122.19, 133.11)	133.68 (124.11, 143.26)	123.26 (115.49, 131.03)	132.12 (119.46, 144.78)	β (95% CI) −11.53 (−24.40, 1.35)	β (95% CI) −7.59 (−24.41, 9.23)
Pulmonary function					β (95% CI)	β (95% CI)
FEV1, L/min	2.91 (2.79, 3.02)	3.05 (2.87, 3.23)	2.85 (2.68, 3.02)	2.90 (2.68, 3.12)	−0.13 (−0.34, 0.09)	−0.26 (−0.54, 0.02)
FEV1, %	98.17 (95.49, 100.85)	103.58 (99.45, 107.71)	97.71 (93.87, 101.56)	91.83 (85.37, 98.28)	−5.36 (−11.89, 1.18)	−10.78 (−19.23, −2.33) *
FVC, L/min	3.77 (3.63, 3.91)	3.99 (3.74, 4.25)	3.67 (3.47, 3.86)	3.74 (3.46, 4.03)	−0.31 (−0.59, −0.02)	−0.46 (−0.86, −0.06) *
FVC, %	103.28 (100.42, 106.15)	111.54 (106.59, 116.49)	102.37 (98.68, 106.06)	94.23 (86.51, 101.94)	−8.44 (−15.44, −1.43) *	−13.73 (−24.05, −3.40) *
DLCO, %	80.43 (77.83, 83.04)	88.98 (84.77, 93.19)	79.68 (76.04, 83.32)	70.46 (65.13, 75.79)	−9.71 (−15.49, −3.92) *	−23.08 (−30.28, −15.89) *
TLC, %	94.05 (91.18, 96.91)	99.2 (94.14, 104.26)	94.31 (90.48, 98.14)	84.38 (77.27, 91.50)	−5.10 (−12.55, 2.34)	−14.19 (−24.40, −3.98) *
RV, %	98.37 (93.83, 102.91)	101.73 (93.31, 110.15)	99.99 (93.20, 106.77)	87.96 (80.00, 95.92)	−3.96 (−15.56, 7.64)	−15.01 (−29.17, −0.85) *
					OR (95% CI)	OR (95% CI)
DLCO < 80%	96 (45%)	15 (27%)	54 (45%)	27 (69%)	2.94 (1.32, 6.51) *	13.02 (0.88, 43.65) *
DLCO < 60%	35 (16%)	2 (4%)	22 (18%)	11 (28%)	4.93 (1.04, 23.31) *	24.47 (3.05, 196.24) *
					β (95% CI)	β (95% CI)
MRC score	1.95 (1.82–2.08)	1.75 (1.54–1.95)	1.96 (1.79–2.14)	2.18 (1.80–2.56)	0.21 (−0.08, 0.50)	0.38 (−0.05, 0.81)
					OR (95% CI)	OR (95% CI)
MRC score (3–5)	48 (24%)	8 (15%)	28 (26%)	12 (31%)	1.89 (0.73, 4.95)	2.54 (0.71, 9.03)
6MWT					β (95% CI)	β (95% CI)
6MWTD (m)	486.90 (471.87, 501.94)	515.33 (489.37, 541.29)	479.65 (457.79, 501.51)	464.65 (431.40, 497.89)	−15.78 (−45.75, 14.20)	−31.59 (−73.19, 10.01)
Percent predicted, male	84.65 (81.55, 87.75)	88.98 (83.74, 94.21)	85.38 (81.07, 89.69)	79.73 (72.69, 86.77)	−3.86 (−10.44, 2.72)	−9.82 (−18.30, −1.33) *
Percent predicted, female	95.97 (91.35, 100.59)	93.09 (85.42, 100.76)	95.00 (88.71, 101.29)	111.58 (94.94, 128.21)	−3.92 (−14.75, 6.91)	5.83 (−8.83, 20.48)
Desaturation (%-point)	2.89 (2.39, 3.39)	2.11 (1.45, 2.77)	3.05 (2.25, 3.85)	3.65 (2.56, 4.73)	0.59 (−0.41, 1.59)	1.39 (−0.00, 2.79)
Borg scale before test	0.79 (0.58, 1.01)	0.57 (0.20, 0.94)	0.85 (0.56, 1.13)	1.02 (0.35, 1.69)	0.17 (−0.36, 0.69)	0.34 (−0.44, 1.12)
Borg scale after test	3.64 (3.24, 4.04)	3.41 (2.67, 4.15)	3.73 (3.19, 4.28)	3.73 (2.67, 4.78)	0.34 (−0.60, 1.29)	0.27 (−1.08, 1.63)
Change in Borg scale	2.82 (2.46, 3.18)	2.83 (2.15, 3.49)	2.85 (2.34, 3.35)	2.69 (1.86, 3.53)	0.13 (−0.68, 0.95)	−0.06 (−1.23, 1.11)
					OR (95% CI)	OR (95% CI)
Desaturation below 92	37 (19%)	6 (11%)	19 (19%)	12 (32%)	1.11 (0.39, 3.19)	3.94 (1.07, 14.48) *
Desaturation ≥4%-point	53 (27%)	12 (22%)	23 (23%)	18 (49%)	0.82 (0.35, 1.91)	3.44 (1.19, 9.91) *
					β (95% CI)	β (95% CI)
Fatigue assessment scale	25.61 (24.29, 26.93)	28.08 (25.45, 30.71)	24.66 (22.81, 26.52)	25.08 (22.42, 27.74)	−1.32 (−4.80, 2.16)	−2.05 (−6.29, 2.19)
					OR (95% CI)	OR (95% CI)
No fatigue	67 (35%)	11 (23%)	42 (40%)	14 (38%)	1.96 (0.86, 4.45)	2.57 (0.86, 7.68)
Mild-to-moderate fatigue	88 (47%)	25 (52%)	45 (43%)	18 (49%)	0.82 (0.41, 1.65)	1.31 (0.52, 3.32)
Severe fatigue	34 (18%)	12 (25%)	17 (16%)	5 (14%)	0.87 (0.34, 2.22)	0.57 (0.16, 2.03)
					β (95% CI)	β (95% CI)
HADS score total	7.94 (6.95, 8.93)	9.60 (7.54, 11.67)	7.59 (6.27, 8.91)	6.76 (4.52, 8.99)	−1.17(−3.79, 1.45)	−2.90 (−6.15, 0.34)
HADS-D score	3.22 (2.71, 3.67)	3.56 (2.53, 4.59)	3.26 (2.59, 3.94)	2.49 (1.59, 3.38)	−0.05 (−1.41, 1.31)	−1.29 (−2.84, 0.26)
HADS-A score	4.75 (4.17, 5.34)	6.04 (4.87, 7.21)	4.32 (3.56, 5.08)	4.27 (2.80, 5.74)	−1.12 (−2.59, 0.35)	−1.61 (−3.53, 0.31)
					OR (95% CI)	OR (95% CI)
HADS-D abnormal score (≥8)	29 (16%)	9 (19%)	16 (16%)	4 (11%)	0.94 (0.35, 2.49)	0.54 (0.14, 2.14)
HADS-A abnormal score (≥8)	43 (23%)	15 (31%)	21 (21%)	7 (19%)	0.73 (0.32)	0.69 (0.22, 2.20)

Data are *n* (%), mean (95% CI). Comparison of groups are presented by regression coefficients (β) (95% CI) for continuous data and as odds ratio (OR) (95% CI) for binary outcomes. * *p* < 0.05. Comparison was made using multivariable analysis including variables age, sex and BMI ** Patients treated at the ward.

**Table 3 jcm-11-02411-t003:** Demographics and outcomes at follow-up stratified for treatment with RaD.

	Oxygen-Dependent Patients Not Receiving Systemic Corticosteroids and Remdesivir (RaD÷) *n* = 52	Oxygen-Dependent Patients Receiving Systemic Corticosteroids and Remdesivir (RaD+) *n* = 86	RaD+ vs. RaD÷ Multivariable Model
			β (95% CI)
Age	63.54 (59.62, 67.45)	61.88 (59.24, 64.53)	−5.27 (−9.33, −1.22) *
Sex			OR (95% CI)
Male	31 (60%)	56 (65%)	1.18 (0.55, 2.51)
Female	21 (40%)	30 (35%)	
Smoking status			OR (95% CI)
Ever	23 (48%)	37 (47%)	1.18 (0.55, 2.51)
Never	25 (52%)	41 (53%)	
			β (95% CI)
Number of comorbidities	1.13 (0.84, 1.43)	1.79 (1.55, 2.03)	0.72 (0.40, 1.04) *
			β (95% CI)
Length of hospital stay	12.27 (8.84, 15.69)	9.22 (7.08, 11.36)	−4.04 (−8.38, 0.29)
			OR (95% CI)
ICU admission	13 (25%)	17 (20%)	0.19 (0.05, 0.74) *
Number of symptoms at follow-up			β (95% CI)
	3.27 (2.49, 4.04)	3.86 (3.15, 4.57)	0.13 (−0.97, 1.24)
			β (95% CI)
MRC score	1.89 (1.64, 2.15)	2.08 (1.85, 2.30)	−0.03 (−0.36, 0.29)
			OR (95% CI)
MRC score 3–5	12 (25%)	22 (28%)	0.81 (0.32, 2.04)
Pulmonary function			β (95% CI)
FEV1, %	101.69 (96.30, 107.07)	95.0 (90.84, 99.16)	−4.17 (−11.11, 2.78)
FVC, %	107.37 (101.08, 113.66)	97.92 (93.89, 101.94)	−6.64 (−13.86, 0.58)
DLCO, %	80.41 (74.79, 86.04)	76.39 (72.21, 80.57)	−2.15 (−8.89, 4.59)
6MWT			β (95% CI)
6MWTD (m)	471.73 (438.67, 504.79)	471.55 (445.56, 497.54)	20.02 (−17.25, 57.31)
Desaturation during 6MWT (%-point)	2.09 (1.25, 2.93)	3.79 (2.76, 4.81)	1.77 (0.27, 3.27) *
Change in Borg scale score			
	2.28 (1.60, 2.95)	3.37 (2.67, 4.07)	1.16 (0.05, 2.27) *
			β (95% CI)
FAS score	23.73 (21.28, 26.19)	25.37 (23.17, 27.57)	0.61 (−2.67, 3.89)
			OR (95% CI)
No fatigue	22 (45%)	28 (38%)	0.78 (0.36, 1.67)
Mild-to-moderate fatigue	23 (47%)	31 (42%)	0.73 (0.34, 1.54)
Severe fatigue	4 (8%)	14 (19%)	1.65 (0.43, 6.19)
			β (95% CI)
HADS score total	5.69 (4.17, 7.21)	8.82 (7.04, 10.60)	2.52 (0.21, 4.83) *
HADS-D score	2.22 (1.52, 2.91)	3.89 (3.00, 4.79)	1.29 (0.20, 2.38) *
HADS-A score	3.47 (2.20, 4.44)	4.93 (3.88, 5.97)	1.23 (−0.22, 2.67)
			OR (95% CI)
HADS-D abnormal score (≥8)	3 (6%)	15 (22%)	3.17 (0.78, 12.94)
HADS-A abnormal score (≥8)	5 (10%)	18 (26%)	2.54 (0.82, 7.84)

Data are *n* (%), mean (95% CI). Comparison of groups are presented by regression coefficients (β) (95% CI) for continuous data and as odds ratio (OR) (95% CI) for binary outcomes. Comparison was made using multivariable analysis including variables age and comorbidities * *p* < 0.05.

## Data Availability

All data used during the study are available on reasonable request.

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
