# Peer review of "Fatigue Is a Major Symptom at COVID-19 Hospitalization Follow-Up"

_jcm, 2022, doi:10.3390/jcm11092411_

Round 1
Reviewer 1 Report
- I am disappointed that the abstract does not state the aim of the study, and limits itself to the conclusion that patients with non respiratory symptoms had more anxiety and depression, and without providing the data and method whixh support that conclusion
- The stated aim in the in the introduction is too vague and the link between aim and methodology is not clear from the text. The main research question to be proven is not clear.
- Recruitment – what was the response rate – was it the same in the three groups. It is not clear how many patient accepted the invitation ?
- The follow up period varied from 3-6 months. Was this something random or structured who to contact, and when ???
- Statistical modelling for binary logistic models, needs to be better explained. Which variables were inserted in the model, how were they chosen ? Was this a stepwise regression model ie excluding non significant models, or not? And if yes was this a forward or a backward stepping. It is impossible for the reviewer/reader to assess the validity of this modelling.
- I note that that the tests/tools/questionnaires were very appropriate what is being measure – but the link aim – method – tool -results – statistical analysis is not well explained.
- Table 1. Is a bit confusing what is in parenthesis – I can see a small asterisk indicates it is a percentage however I would include the “%” once the first line is a confidence interval and not a percentage.
- In my opinion the 3 groups are not comparable by a univariate analysis ie fisher or , because clearly group one is younger with fewer co-morbodities and more females Table 1. Is of value for describing the patients, but is very difficult or impossible to prove the research question by a simple comparison.
- Table 2. Following question on multivariate modelling the reviewer cannot know what the odds ration quoted is a predictor of
- Data that patients needing oxygen had worse outcomes in lung function, but to show it is from covid a multivariate model on outcome either of a binary or ordinal nature is necessary because they are not matched.
- I am not sure that the conclusions mentioned in the discussions are justified by the data, nor are they declared in the aims or the research question. In my idea the groups are not well matched, and the multivariate analysis unclear
Author Response
Please see the attahment

Reviewer 2 Report
This is a very well written manuscript, very well organized, overall very clear. Strengths: Fatigue was measured in the FAS and scored by standardized PROM. Multicenter observational study. Weakness: small population, numbers even lower when analyzed in subgroups, no data from pre-hospitalization.
I have several recommendations that would make the article stronger:
Title: I would recommend to strongly considering changing the title to better represent the results; mild disease makes one think of out of the hospital, asymptomatic or minimally symptomatic covid patients; once hospitalized, the disease is no longer considered mild. Could consider something more impactful like: “Fatigue is a major symptom at COVID-19 hospitalization follow up “ or change to Chronic symptoms after covid hospitalization to include all your findings.
Abstract emphasizes multiple chronic post covid symptoms, while the title and the manuscript emphasize the fatigue. Please clarify which direction should the title and manuscript go towards.
Methods: please clarify the time frame form covid to data collection; clarify if data collection was obtained in clinic at follow up. This was done in the abstract but I cannot not find it in the manuscript.
Methods/results:
There are several other analysis, not presented in the manuscript that I would be interested in seeing in the methods/results section, could be a table summarizing these analysis: analysis related to FAS and DLCO, analysis related to FAS and Anxiety/Depression, FAS and smoking status, FAS and dyspnea, FAS and anticoagulation, and FAS and 6 mwt distance. I suspect that there is no data available related to Anxiety/Depression in this cohort prior to covid, but that would also be of interest if data is available. Based on these results, if new statistically significant results are found, could consider a multivariate analysis to evaluate which variable has the larger impact on FAS.
The analysis on FAS and RaD treatment could also be expanded to include all patients (Fas in RaD treatment vs no RaD treatment).
It is beyond the scope of this manuscript, but would be interesting to see the correlation between RaD treatment and post covid DLCO levels.
Also if available, would be interesting to see what the presenting symptom is for patients requiring hospitalization in group 1 and if Fatigue was documented in the chart as a symptom.
Discussion:
Would start the discussion with the first sentence in paragraph 3 and emphasize more that most patients reported fatigue and FAS scores were increased in the majority of patients. “Fatigue was a major symptom in our study population and the incidence at follow- up was in line with several previous studies reporting fatigue in 52-63% of participants.”
Would limit a little bit the discussion on other symptoms at follow up: dyspnea, MRC, depression, anxiety or emphasis their potential implications in the findings related to fatigue.
Author Response
Please see the attahment

Round 2
Reviewer 1 Report
The paper is greatly approved however it would have been more robust if a multivariate model, preferably a stepwise logistic regression in comparing group 2 with one and group 3 with one. A univariate analysis does not distinguish between dependent and independent variables
A stepwise binary regression model eleminates non-significant variables and increases the stastistical power of the test and may detect more predictors.
Would suggest review of this fact by an independent biostatistician before publication
Furthermore since the tables are rather heavy - fisher plot of the odds ration with the y axis at number one would make it much easier for readers to interpret the data
example follow link https://stackoverflow.com/questions/47085514/simple-way-to-visualise-odds-ratios-in-r
Author Response
On behalf of my co-authors, I am pleased to submit a revised manuscript.
Once again we thank the reviewer for the supplementary comments. We have provided a point-by-point response to each comment in the table attached.

Reviewer 2 Report
Abstract: would recommend to clarify aims to include other symptoms reported in addition to fatigue ( depression, anxiety, exercise capacity..etc) and pulmonary function tests. This is a nicely done study and having a good comprehensive abstract will increase the readers interest and the chance that will be cited in the future. The data available in the manuscript related to anxiety and depression is also relevant, in particular the correlation with Fatigue and I would consider mentioning this in the abstract.
Thank you for performing the extra analyses on Fatigue, I think it makes the manuscript stronger.
On discussion, line 276 would consider deleting fatigue and listing other symptoms instead.
Would add a couple more sentences in the discussion on relationship between FAS and HADS and FAS and 6mwt. How would you explain these findings?
Author Response

(The authors gave the same response as above.)
